# TGF-β Pathways Stratify Colorectal Cancer into Two Subtypes with Distinct Cartilage Oligomeric Matrix Protein (COMP) Expression-Related Characteristics

**DOI:** 10.3390/biom12121877

**Published:** 2022-12-14

**Authors:** Jia-Tong Ding, Hao-Nan Zhou, Ying-Feng Huang, Jie Peng, Hao-Yu Huang, Hao Yi, Zhen Zong, Zhi-Kun Ning

**Affiliations:** 1Department of Gastrointestinal Surgery, The Second Affiliated Hospital of Nanchang University, Nanchang 330006, China; 2The Second Clinical Medicine School, Nanchang University, Nanchang 330006, China; 3Queen Mary School, Nanchang University, Nanchang 330006, China; 4Department of Day Ward, The First Affiliated Hospital of Nanchang University, Nanchang 330006, China

**Keywords:** colorectal cancer, COMP, TGF-β

## Abstract

Background: Colorectal cancers (CRCs) continue to be the leading cause of cancer-related deaths worldwide. The exact landscape of the molecular features of TGF-β pathway-inducing CRCs remains uncharacterized. Methods: Unsupervised hierarchical clustering was performed to stratify samples into two clusters based on the differences in TGF-β pathways. Weighted gene co-expression network analysis was applied to identify the key gene modules mediating the different characteristics between two subtypes. An algorithm integrating the least absolute shrinkage and selection operator (LASSO), XGBoost, and random forest regression was performed to narrow down the candidate genes. Further bioinformatic analyses were performed focusing on COMP-related immune infiltration and functions. Results: The integrated machine learning algorithm identified COMP as the hub gene, which exhibited a significant predictive value for two subtypes with an area under the curve (AUC) value equaling 0.91. Further bioinformatic analysis revealed that COMP was significantly upregulated in various cancers, especially in advanced CRCs, and regulated the immune infiltration, especially M2 macrophages and cancer-associated fibroblasts in CRCs. Conclusions: Comprehensive immune analysis and experimental validation demonstrate that COMP is a reliable signature for subtype prediction. Our results could provide a new point for TGFβ-targeted anticancer drugs and contribute to guiding clinical decision making for CRC patients.

## 1. Introduction

Colorectal cancer (CRC) is one of the leading causes of newly diagnosed cancer cases and cancer-related deaths worldwide [1]. Characterized as a highly heterogeneous disease, CRC is classified into four consensus molecular subtypes (CMS) based on bulk transcriptomics [2,3]. Among these subtypes, prominent activation of transforming growth factor-β (TGF-β) constitutes one of the most important features of the CMS4 subtype (mesenchymal, 23%), which is accompanied by stromal invasion, epithelial-to-mesenchymal transition (EMT), and angiogenesis [3,4]. Compared with the other three CMS groups, CMS4 tumors have a higher tendency of long-term relapse and a low response to chemotherapy, leading to the poorest prognosis [4]. Thus, a clearer landscape of the molecular features of CMS4 CRCs will provide individualized treatments for patients accurately and show great value in clinic application.

Alterations of cytokine/chemokine networks (for example, TGF-β) in the tumor microenvironment (TME) constitute one of the most crucial pathways involved in cancer tumorigenesis via regulating signaling pathways in cancerous cells directly and the tumor–stromal interaction [5,6]. TGF-β regulates cell proliferation, migration, and differentiation, serving as a tumor suppressor in the early stages of tumorigenesis but as a tumor promotor at later stages [7]. Another alteration in the TME induced by TGF-β activation is changes in several stromal cells, including cancer-associated fibroblasts, natural killer cells, tumor-associated neutrophils, tumor-associated macrophages, naïve CD4+ T cells, type 1 T helper cells, and regulatory T lymphocytes [6]. Based on the above-mentioned features, TGF-β-targeting agents have been identified as promising anticancer drugs, but their clinical translation is limited due to the unduly multifunctional roles of TGF-β in cancerous and normal cells [8]. Thus, more studies on this topic are needed to obtain a more comprehensive and accurate understanding of TGF-β for developing precisely targeted anticancer drugs.

Cartilage oligomeric matrix protein (COMP), namely thrombospondin-5 (TSP-5), interacts with type I and type II collagen to regulate the assembly and stabilization of the extracellular matrix [9,10]. With a pentameric structure composed of five identical monomers, COMP can simultaneously interact with multiple TGF-β1 molecules, contributing to the high efficacy in the sustained activation of the TGF-β signaling pathway [11]. Moreover, COMP is significantly upregulated in CRCs and is co-expressed with key EMT genes and associated with poor survival [10]. Although the crosstalk of COMP and TGF-β is implicated in various pathophysiological processes, such as collagen secretion and pancreatic tumor progression [12,13], its specific roles in CRCs remain vague; thus, a general analysis is indispensable and urgently required.

In this study, we identified two TGF-β pathway subtypes in CRCs. With weighted correlation network analysis (WGCNA) and machine learning algorithms, we ultimately identified COMP as the hub gene that mediates the difference between the two subtypes. Correspondingly, validation tests including the receiver operating characteristic (ROC) were conducted, which demonstrated an excellent predictive power of COMP for C1 and C2 subtypes. This study provides more realistic information and a more convenient classification system for clinicopathological features, which can serve as a guide for clinical decision making. Furthermore, this study aims to broaden our view on the development of TGF-β-targeting anticancer agents and to improve CRC patients’ survival.

## 2. Materials and Methods

### 2.1. Data Retrieval and Preprocessing

The molecular data (RNA sequencing, DNA Methylation, somatic mutation, and copy number alteration) of colorectal cancer and paracancerous tissue were downloaded from the TCGA database. Duplicate cases and patients treated by chemotherapy or radiation therapy were regarded as invalid and removed, and 455 samples were then obtained for the following analyses. In addition, RNA-seq data of normal intestinal tissue were downloaded from the GTEx database. All sequencing data were converted to TPM format.

### 2.2. Consensus Clustering and Weighted Gene Co-Expression Network Analysis

To cluster CRC samples based on TGF-β-related pathways, the Molecular Signatures Database (www.gsea-msigdb.org/gsea/msigdb, accessed on 28 May 2022) was applied to download 5 gene sets of TGF-β-related pathways (BIOCARTA TGFB PATHWAY, REACTOME SIGNALING BY TGFB FAMILY MEMBERS, HALLMARK TGF BETA SIGNALING, WP TGFBETA SIGNALING PATHWAY, KEGG TGF BETA SIGNALING PATHWAY) [14]. Differential genes between normal and cancerous tissues were selected by the T-test and Wilcoxon test, respectively. Further feature selection was performed based on the Cox regression model. Consensus clustering identified two distinct TGF-β-related patterns via the method in the “CancerSubtypes” R package.

A network of gene co-expression was constructed to identify the TGF-β cluster-related module. After filtering out genes with the top 5000 median absolute deviations, we calculated the connection strength and established a scale-free network. Then, we examined the scale independence and average connectivity degree of modules using the gradient method. Next, the degree of independence was set as 3, the most suitable power value, and then scale-free gene co-expression networks were generated. Finally, the 3 modules that had the highest coefficient of correlation with TGF-β clusters were identified as the key modules.

### 2.3. Functional and Pathway Enrichment Analysis

Among the “clusterProfiler” R package, Gene Ontology (GO) analysis and Kyoto Encyclopedia of Genes and Genomes (KEGG) pathway analysis were used to further elucidate the biological functions and signaling pathways of the identified genes in the key modules.

### 2.4. Feature Selection with Machine Learning Algorithms

Candidate hub genes were then obtained via dimension reduction approaches, including XGBoost, lasso regression, and random forest regression, which were induced to improve the prognostic signature accessibility of the prognostic genes. After integrating the multiple algorithms, COMP was eventually identified as the TGF-β-related hub gene.

### 2.5. Association between COMP and Clinical Features

After downloading the clinical features of CRC from TCGA, we compared the COMP expression level among patients with different ages, survival events, TNM, and pathologic stages. In addition, the ROC plotter tool (http://www.rocplot.org/, accessed on 26 August 2022) was utilized to analyze the association between COMP expression level and response to therapy based on the transcriptome-level data from patients with CRC [15].

### 2.6. Tumor Immune Estimation Resource

Immunogenomic analysis was performed by our ImmuCellAI algorithm with 24 immune cells to analyze the correlation of immune infiltration with COMP expression and methylation in pan cancers. For CRCs, the ssGSEA algorithm from the R package “GSVA” was conducted to demonstrate immune infiltration differences between tumors with low and high COMP expression levels and the correlation between COMP and the immune infiltration of 24 immune cells.

### 2.7. Analysis of COMP Expression Correlation with Immunological Characteristics of TME

Using SangerBox (http://vip.sangerbox.com/home.html, accessed on 28 August 2022), we analyzed the correlation between COMP and immunomodulators, including chemokines, receptors, MHCs, immunoinbibitors, and immunostimulators [16]. After identifying the co-expressed genes in CRCs with a Spearman’s correlation coefficient greater than 0.8 using cBioPortal (https://www.cbioportal.org/, accessed on 27 August 2022), we ranked the gene prioritization of these genes across four immunosuppressive parameters, including T cell exclusion score, T cell dysfunction score, response to immune checkpoint blockade (ICB) therapy, and gene knockout phenotype, in CRISPR screens with the regulator prioritization module of the TIDE algorithm (http://tide.dfci.harvard.edu/, accessed on 27 August 2022) [17,18]. The T cell dysfunction score hinted at the effects on cytotoxic T cells, and the z-score in the Cox PH regression represented the role of gene expression on patient survival in ICB treatment cohorts. The normalized logFC in CRISPR screens was used for evaluating the effect of gene knockout-mediated and lymphocyte-induced tumor death in cancer models. The T cell exclusion score reflected the collective roles of the T cell exclusion of three immunosuppressive cell types, including CAFs, MDSCs, and M2-TAMs [19].

### 2.8. Interaction Network and Enrichment Analysis

The STRING website (https://cn.string-db.org/, accessed on 28 August 2022) was employed to identify the network interactions of the top 50 related genes of COMP in CRCs from cBioPortal, which was then visualized with Cytoscape software (version 3.7.2). GO and KEGG pathway analysis was performed to explore the biological functions and signaling pathways of the co-expressed genes with a Spearman’s correlation coefficient greater than 0.6. With Gene Set Enrichment Analysis (GSEA), gene expression profile data of READ (rectum adenocarcinoma) and COAD (colon adenocarcinoma) were obtained from TCGA and then divided into a high expression group (≥50%) and a low expression group (<50%) according to the expression level of COMP. A *p*-value < 0.05 (as needed) and an FDR <0.25 (as needed) were considered statistically significant [20].

### 2.9. Statistical Analysis

Correlation coefficients between variables were computed using Pearson and Spearman correlation analyses. The *T*-test and Mann–Whitney U test were applied to calculate the differences in continuous variables. The risk score toward the prognostic significance was compared using the log-rank test. A validation test for the prediction of TGF-β subtypes with COMP was performed via the receiver operating characteristic (ROC) curves. All statistical tests which utilized R software (version 4.2.0) followed the two-sided principle, and statistical significance was determined as *p* < 0.05.

## 3. Results

### 3.1. TGF-β Pathways Stratify CRC into Two Subtypes

In this study, 258 genes were systematically identified from five gene sets of TGF-β-related pathways; then, only 34 overlapping genes passed the feature selection based on the Cox regression model. After clustering CRC patients into C1 and C2 subtypes, we comprehensively analyzed the difference of immune checkpoint-related genes in mRNA expression, DNA methylation, gene amplification frequency, and gene deletion frequency between the two subtypes. The results indicate that gene amplification and deletion frequency were positively correlated with C1 but negatively correlated with C2 in almost all experimental types (Figure 1A). Moreover, to further validate the clinical relevance of the different subtypes, we analyzed the cumulative survival curve and found that the C2 group was linked to the favorable survival of the patients (*p* < 0.001) (Figure 1B). Considering the differences in several aspects mentioned above, WGCNA was performed to explore the key gene modules. With the soft domain value set to 3, R^2^ was greater than 0.8, suggesting that the data conformed to a power law distribution and were suitable for subsequent analysis. Moreover, the mean connectivity tended to be stable, suggesting that, when the soft domain value was further increased, the effect on the results was not significant (Figure 1C). Furthermore, with WGCNA, we obtained 12 non-gray gene modules, among which the blue, magenta, and turquoise modules had the strongest positive correlation with the C1 subtype and strongest negative correlation with the C2 subtype (Cor = 0.5 and *p* < 0.001), suggesting that these three module genes were most closely related to the differences between the two subtypes. Multivariate regression analysis was performed to identify the C1 and C2 subtypes as independent risk factors (Table 1).

### 3.2. GO and KEGG Enrichment Analysis for Related Gene Modules

Due to the strongest correlation between blue, magenta, and turquoise modules and the two subtypes, GO and KEGG enrichment analysis was used to explore the function of the significantly altered genes. The magenta module was mainly enriched in muscle contraction, the muscle system process, and myofibril assembly. The turquoise module was mainly enriched in cell–substrate adhesion, epithelial cell proliferation, and ameboidal-type cell migration. The blue module was mainly enriched in the extracellular matrix organization, extracellular structure organization, and external encapsulating structure organization (Figure 2A). As for KEGG analysis, the magenta module was significantly enriched in vascular smooth muscle contraction, hypertrophic cardiomyopathy, and dilated cardiomyopathy; the turquoise module was significantly enriched in the PI3K/Akt signaling pathway, focal adhesion, and ECM–receptor interaction; and the blue module was significantly enriched in human papillomavirus infection, PI3K/Akt signaling pathway, and protein digestion and absorption (Figure 2B). Intriguingly, the blue module and the turquoise module both upregulated in the PI3K/Akt signaling pathway.

### 3.3. Machine Learning Methods Identified COMP as the Hub Gene

LASSO, XGBoost, and random forest regression (RFR) were selected to explore the key genes mediating the differences between the two subtypes. After normalizing and integrating the output, COMP was identified as the hub gene (Figure 3A). Further bioinformatic analysis found that the expression of COMP was higher in the C1 group than in the C2 group (*p* < 0.05) (Figure 3B). Receiver operating characteristic (ROC) curve analysis indicated the excellent performance of COMP in predicting the subtype of TGF-β (AUC = 0.91) (Figure 3C). The Sankey diagram demonstrates the CMS tumor type composition and COMP expression level in C1 and C2 clusters, with all CMS4 subtypes being clustered as C1 (Figure 3D).

### 3.4. Association between COMP and CRC Clinical Features

CRC cases with complete observation data from TCGA were divided into low and high expression groups according to the median COMP expression level. Statistical analysis exhibited a significant association between high COMP expression and advanced T, N, and pathologic stages (Figure 4A,C,H), worse OS (overall survival), PFI (progression-free interval), and DSS (disease-free survival) events (Figure 4D–F), and patients under the age of 65 (Figure 4G), but there existed no significant correlation between COMP expression and different M stages (Figure 4B). Moreover, the low mRNA expression level of COMP was significantly associated with resistance to chemotherapy in the CRC cohorts (Figure 4I). Similarly, a low COMP expression level may represent resistance to immune checkpoint inhibitor therapy (Figure 4J).

### 3.5. COMP Expression Level and Immune Infiltration Analysis in Pan Cancers

Pan-cancer analysis revealed that COMP was highly expressed in various cancers, such as BLCA, BRCA, CESC, CHOL, COAD, ECSC, HNSC, LIHC, LUAD, LUSC, PAAD, READ, SKCM, STAD, THCA, and UCEC (Figure 5A). More importantly, COMP exhibited excellent diagnostic values for CRCs with an AUC equaling 0.932 and a 95% confidence interval equaling 0.908–0.955 (Figure 5B). In COAD and READ, the expression of COMP steadily increased as the tumor progressed (Figure 5C, D). Western blot analysis of COMP and TGF-β1 protein expression levels suggested that TGF-β1, as well as COMP, were upregulated in colorectal cancer cell lines compared with the normal colonic cell line (NCM-460) (Appendix A). Immune infiltration analysis exhibited a significantly positive correlation between COMP expression and the infiltration score in most cancers, with COMP positively correlating with CD4+ T cells, naïve CD8 cells, iTreg, macrophages, Tfh, Th2, and Tr1, but negatively correlating with neutrophils in most cancers (Figure 5E). The further analysis between the DNA methylation levels of COMP and tumor-infiltrating immune cells indicated positive correlations with CD4+ T cells, naïve CD8 cells, effector memory, monocytes, and nTreg, Th1, and Tr1 in COAD, but no significant correlations in READ. The pan-cancer analysis also revealed a positive correlation between the DNA methylation level of COMP and the neutrophils that existed in most cancers (Figure 5F).

### 3.6. Correlation between COMP and Immune Infiltration in CRC

Detailed immune infiltration analysis of CRCs revealed that a high COMP expression level was related to the higher enrichment score of 19 immune cells, but the lower enrichment scores of Th17 and Th2 cells. However, NK CD56 bright cells, T helper cells, and Tcm exhibit no significant difference between the two groups with low and high COMP expression divided by the median (Figure 6A). Correspondingly, the correlation analysis indicated that COMP negatively associated with Th17 and Th2 cells, but positively with most infiltrating immune cells, especially NK cells, macrophages, iDC, and mast cells, with a Spearman’s correlation coefficient greater than 0.4 (Figure 6B,C and Appendix A). Considering the two subtypes of macrophages in the TME [21], we also analyzed the correlation between COMP and the biomarkers of M1 and M2 macrophages. The results indicate a stronger association between COMP and M2 than M1 macrophages, with a significantly positive correlation between COMP and the biomarkers of M2 macrophages, including CD163 (r = 0.477, *p* < 0.001), MS4A4A (r = 0.440, *p* < 0.001), and VSIG4 (r = 0.503, *p* < 0.001) (Figure 6D), but a poorer correlation with the biomarkers of M1 macrophages, including IRF5 (r = 0.338, *p* < 0.001), NOS2 (r = −0.284, *p* < 0.001), and TGS2 (r = 0.007, *p* > 0.05) (Figure 6E).

### 3.7. Immunomodulatory Relevance of COMP

The immunomodulatory role of COMP was depicted via the associations between the COMP expression level and immunomodulators. The findings reveal that COMP was positively correlated with a majority of immune inhibitors and stimulators in several cancers, with TNFSF4, an immune stimulator, and HAVCR2, an immune inhibitor, exhibiting the strongest correlation with COMP (Figure 7A). Moreover, significant correlations were found between COMP expression and the expression of immune checkpoint genes in most cancers. In CRCs, COMP showed a highly significant positive correlation with most immune checkpoint genes, while exhibiting negative correlations with several chemokines, including CCL20, CXCL2, CXCL1, CXCL3, and CXCL17 (Figure 7B). Considering the synergy between related genes, we identified the co-expressed genes of COMP in CRCs and exhibited the most related genes with a Spearman’s correlation coefficient greater than 0.8 (Figure 7C and Appendix A). Then, the association between each gene and four immunosuppressive indices was summarized in a range of cohorts. Among the three cell types promoting T cell exclusion, only cancer-associated fibroblasts were positively associated with the expression levels of both COMP and its co-expressed genes (Figure 7D). The significantly positive correlation between COMP and cancer-associated fibroblasts (CAFs) was validated in pan cancers (Figure 7E) and CRCs (Figure 7F) with TIMER2.0 (http://timer.comp-genomics.org/, accessed on 31 August 2022) [22]. Above all, these results suggest that COMP may provide some theoretical support for tumor immunotherapy.

### 3.8. Enrichment Analysis of COMP-Related Genes

The top 50 COMP-related genes were selected, and their interactive relationship was analyzed and visualized with STRING and Cytoscape, respectively (Figure 8A). Then, GO and KEGG enrichment analysis was performed for the COMP-related genes with a Spearman’s correlation coefficient greater than 0.6. As expected, the results suggest that COMP-related genes participated in the TGF-β signaling pathway and ECM–receptor interaction (Figure 8B and Appendix A). Further GSEA analysis indicated that the TGF-β signaling pathway and cancer-related pathway were significantly activated in CRC tissues with high COMP expression levels compared with the low COMP expression group. Interestingly, the high COMP expression level was related to the regulation of endothelial migration in COAD but mitochondrial electron transport in READ (Figure 8C).

## 4. Discussion

CRC is a highly heterogeneous malignant tumor containing at least four consensus molecular subtypes [3]. Featuring the prominent activation of TGF-β signaling, the CMS4 subtype has the lowest survival rate and most aggressive invasiveness, suggesting the potential role of TGF-β pathways in CRC progression [23,24,25]. Our previous study indicated that accurate classification helps conduct reliable subtype-specific prognostic signatures, contributing to more precise clinical decision making and immunotherapeutic strategies [26]. Continuing this line of thinking, the present study stratified CRC into two subtypes, focusing on the associations between TGF-β pathways and CRC clinicopathologic features. With WGCNA and machine learning analysis, we identified COMP as the most critical gene mediating the differences between the two CRC subtypes. Further bioinformatics analysis revealed the influence of COMP on immune cell infiltration. These findings fill in gaps in the field regarding the specific crosstalk of COMP and TGF-β pathways and offer a new approach for clinical decision making and immunotherapeutic strategies to improve CRC patient prognosis and risk stratification.

As a cytokine with dichotomous roles, TGF-β inhibits proliferation and promotes cell cycle arrest and apoptosis in normal and premalignant cancer cells but promotes tumorigenesis and metastasis in late-stage cancerous cells [27]. Increased TGF-β in the tumor microenvironment predicts adverse outcomes in CRC patients, partly due to the TGF-β-induced immune evasion mechanism that promotes T cell exclusion and blocks the acquisition of the Th1 effector phenotype [28]. Consistently, obtained with unsupervised hierarchical clustering based on the TGF-β signaling pathway, the two subtypes of CRC in the present study exhibited significant differences in survival outcomes and the expression, methylation modification, amplification, and deletion frequency of immune checkpoint-related genes. Extracellular matrix (ECM) organization and ECM–receptor interaction are considered the main indicators of the differences. ECM, more often named TME, serves as a key determinant in tumor progression, metastasis, and prognosis [29]. After normal cells suffer damage, the development, invasion, and metastasis of cancerous cells fail to occur without support from the ECM [30]. With the tumor progressing, the normally organized and strictly controlled ECM is reorganized and becomes irregular in density, composition, and structure, leading to dysregulated cellular functions and tumor progression in CRC [31,32]. The interactions of cellular receptors and the ECM promote the development of EMT in cancerous cells and play an important role in CRC progression and metastasis [33,34]. In the process of ECM transformation, the activation of TGF-β signaling is associated with upregulated ECM-related genes in cancer and promotes a transition from fibroblasts to myofibroblasts or cancer-associated fibroblasts, leading to enhanced ECM component accumulation and physical forces to stiffen the ECM [35,36]. Moreover, TGF-β mediates tumor angiogenesis directly or indirectly and induces the formation of an immunosuppressive tumor microenvironment [37]. Hence, through adequately exploring the association of TGF-β signaling-related TME alterations and CRC progression, it is not surprising that C1 and C2 subtypes exhibited significant differences in survival outcomes and the status of immune checkpoint genes.

COMP was identified as the most crucial hub gene that represented the differences between C1 and C2 subtypes with machine learning analysis. Moreover, COMP exhibited a significant expression difference between the two subtypes and a relative classification performance with AUC equaling 0.91, suggesting a significant correlation between the levels of expression of COMP and TGF-β, which is consistent with previous studies [38,39]. Moreover, Zhong et al. reported that COMP promoted EMT and malignant progression and was highly upregulated in tumors, especially in highly malignant CRCs [40]. Further bioinformatic analysis revealed that the COMP expression level was significantly positively correlated with NK cells, iDC, mast cells, CAFs, and macrophages, especially M2 macrophages, in CRC. Macrophages are one of the most common non-cancerous cells in the colorectal cancer microenvironment [41]. A high COMP level was related to a high risk for prostate cancer patients with a higher fraction of regulatory T cells and M2 macrophages [42]. The polarization state of macrophages, rather than their overall density, was associated with cancer-specific survival, with M1-like macrophages having anti-tumor effects but M2-like macrophages performing pro-tumorigenic roles [43,44]. Liu et al. reported that CAF-derived COMP contributed to EMT and cancer stemness in hepatocellular carcinoma [45]. Regulated by the TGF-β signaling pathway, CAFs promote the exclusion of CD8+ T cells from the tumor mass, which leads to the failure of most anti-cancer immunotherapy [46,47,48]. Based on the above statements and comprehensive validations, we suggest that the novel TGF-β pathway clusters which were closed in conjunction with COMP in our study are reliable and have potential for leading novel immunological therapies and improving CRC patient prognosis.

In our study, GSEA suggested the functions of COMP differed in COAD and READ, which may be related to the different pathological characteristics of the two tumors. A high COMP expression level was related to endothelial migration in COAD, but in READ, the high COMP expression level was correlated with mitochondrial electron transport, including “MITOCHONDRIAL ELECTRON TRANSPORT NADH TO UBIQUINONE”, “RESPIRATORY ELECTRON TRANSPORT CHAIN”, and “ATP SYNTHESIS COUPLED ELECTRON TRANSPORT”. Due to the multifaceted functions involved in cell cycle progression and survival, mitochondria have received increasing attention during tumorigenesis, including in CRC [49]. Genes related to mitochondrial function have been associated with tumorigenesis and tumor localization [50]. For example, ubiquinol-cytochrome c reductase binding protein (UQCRB), the crucial regulator for mitochondrial complex III stability and electron transport, was upregulated in CRC tissues, and high UQCRB expression exhibited in most READ samples (5/5 patients, 100%) [51]. During TGF-β-induced EMT, ROS production was stimulated from mitochondria and NADPH oxidase 4 (NOX4), leading to a more oxidative intracellular environment and, subsequently, a more permissive state for EMT [52]. Thus, the COMP-related mitochondrial electron transport process might participate in TGF-β-induced redox imbalance, which may partly explain the higher rates of pulmonary metastasis in rectal cancer patients than colon cancer patients [53]. Our findings could provide essential mechanism complements for the emerging precise treatment toward COMP. Above all, COMP was a candidate molecule for distinguishing rectal cancer from colon cancer, and regulating COMP expression may serve as a cancer-targeting strategy.

Several limitations in this research should be addressed. First, more databases of patients from different regions should be included for more independent external validations. Second, the study only included the samples from the tumor core; however, there are potential differences in the microenvironment features of distinct tumor spatial regions. Therefore, more multicenter, prospective, and well-designed studies are needed.

## Figures and Tables

**Figure 1 biomolecules-12-01877-f001:**
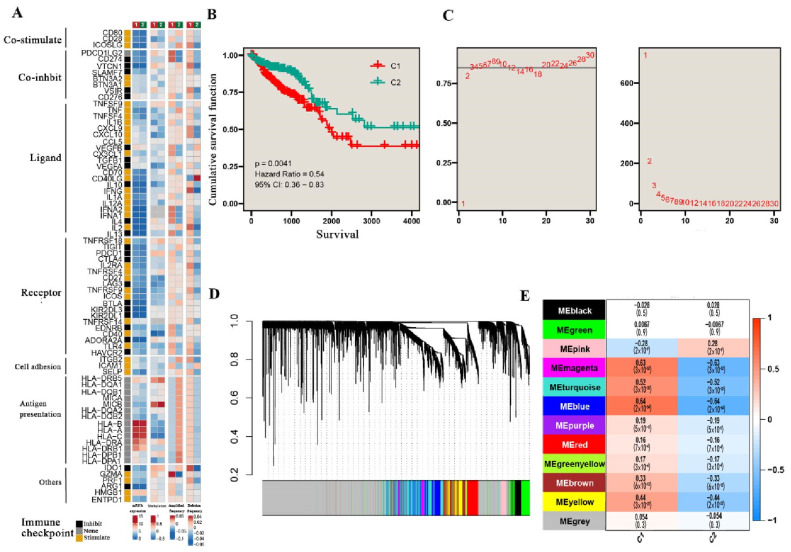
The differences between two subtypes of CRC. (**A**) The comparison of the immune checkpoint-related gene expression, methylation, and mutation correlation between C1 and C2 subtypes. (**B**) Survival outcome differences between two subtypes. (**C**) The soft domain value and mean connectivity setting for WGCNA. (**D**) Cluster dendrogram of WGCNA. (**E**) The correlation between gene modules and subtypes.

**Figure 2 biomolecules-12-01877-f002:**
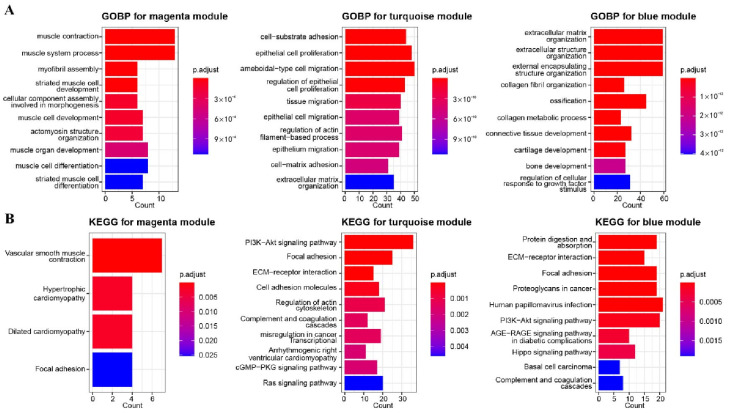
Enrichment analysis of related gene modules. (**A**) GOBP analysis for magenta, turquoise, and blue modules. (**B**) KEGG analysis for magenta, turquoise, and blue modules.

**Figure 3 biomolecules-12-01877-f003:**
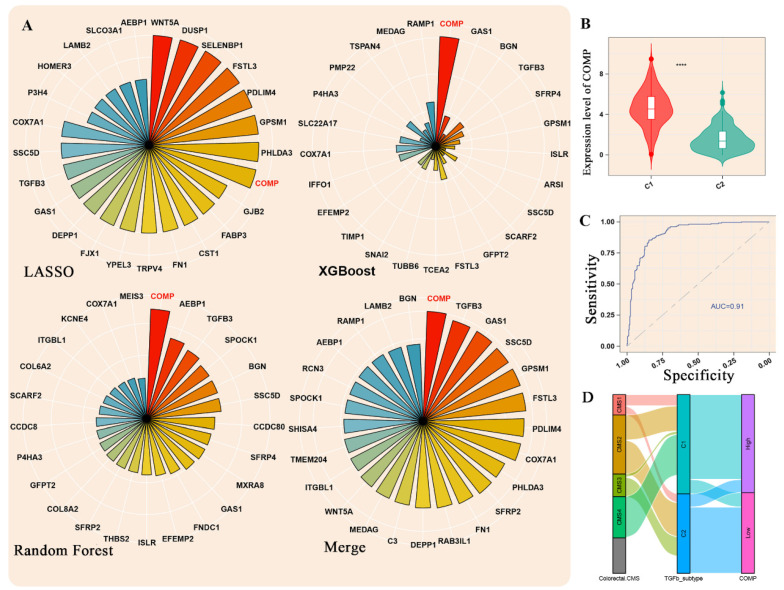
Identification of the hub gene mediating the differences between C1 and C2 subtypes. (**A**) LASSO, XGBoost, random forest, and an integrated algorithm were performed to explore the most important hub gene. (**B**) The expression level of COMP, the hub gene, between two groups. **** *p* < 0.0001. (**C**) ROC exhibited an excellent classification role of COMP for C1 and C2, with AUC equaling 0.91. (**D**) The Sankey diagram shows the relationship between CMS, TGF-β subtype, and COMP expression level.

**Figure 4 biomolecules-12-01877-f004:**
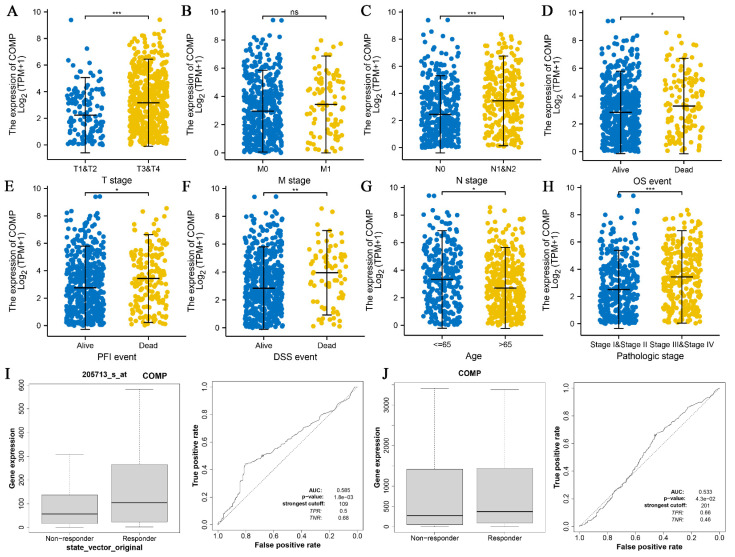
The correlation between COMP and clinical features of CRC patients. The correlation between COMP and (**A**) T stage, (**B**) M stage, (**C**) N stage, (**D**) overall survival (OS) event, (**E**) progression-free interval (PFI) event, (**F**) disease-specific survival (DSS) event, (**G**) patient age, (**H**) pathologic stage, (**I**) response to chemotherapy, and (**J**) response to immunotherapy. * *p* < 0.05, ** *p* < 0.01, *** *p* < 0.001.

**Figure 5 biomolecules-12-01877-f005:**
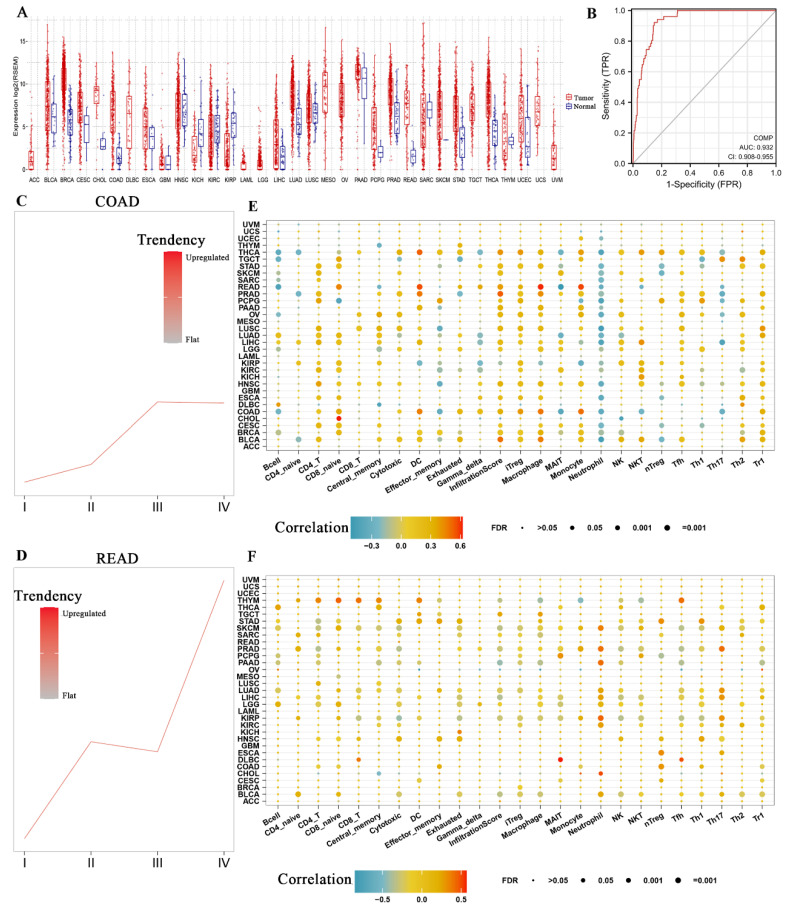
The expression differences of COMP in cancer and the correlation with immune infiltration. (**A**) A pan-cancer analysis for the expression differences of COMP between tumor and normal tissues. (**B**) ROC exhibited an excellent diagnostic role of COMP for CRCs with an AUC equaling 0.932. (**C**) COMP was upregulated in advanced COAD. (**D**) COMP was upregulated in advanced READ. (**E**) The correlation between COMP and immune infiltration in pan cancers. (**F**) The correlation between COMP methylation and immune infiltration in pan cancers.

**Figure 6 biomolecules-12-01877-f006:**
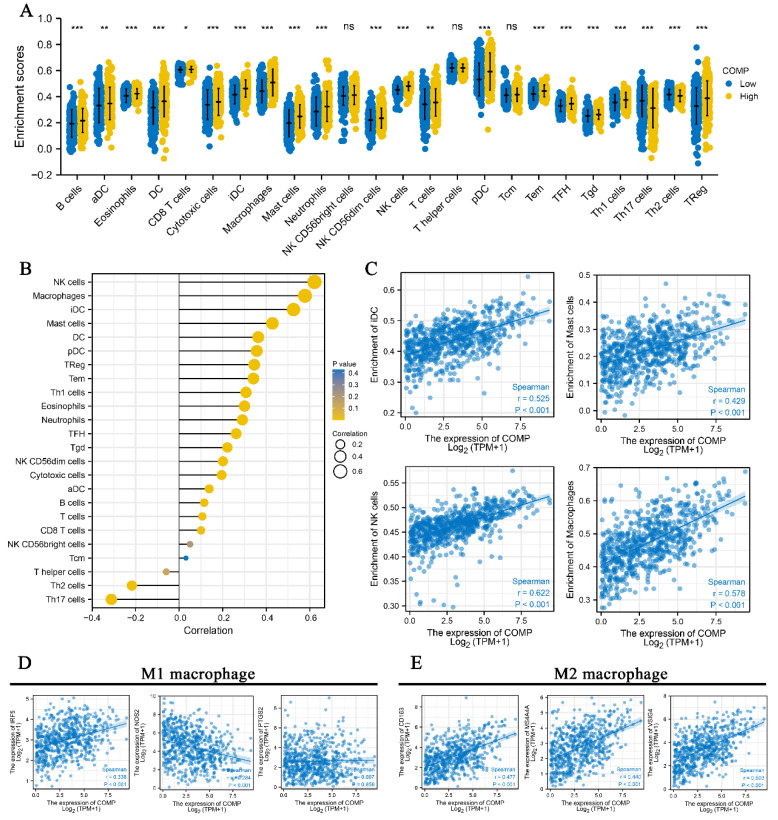
The correlation between COMP and immune infiltration in CRCs. (**A**) The immune infiltration difference between patients with low and high COMP expression levels. (**B**) The correlation between COMP and immune infiltration of 24 immune cells. (**C**) The correlation between COMP and immune infiltration of iDC, NK, mast cells, and macrophages. (**D**) The correlation between COMP and biomarkers of M1 macrophages. (**E**) The correlation between COMP and biomarkers of M2 macrophages. * *p* < 0.05, ** *p* < 0.01, *** *p* < 0.001.

**Figure 7 biomolecules-12-01877-f007:**
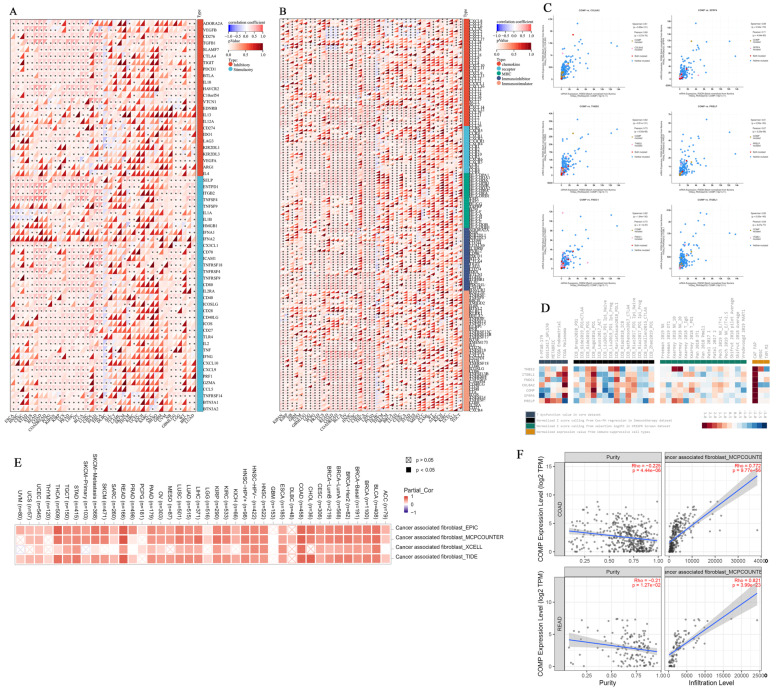
The impact of COMP in immune regulators, immune checkpoint-related molecules, and tumor immune microenvironments. (**A**) The correlation between COMP and immune regulators. (**B**) The correlation between COMP and immune checkpoint-related genes. (**C**) The co-expressed genes of COMP with Spearman’s correlation coefficient greater than 0.8. (**D**) The association between COMP-related genes and four immunosuppressive indices. (**E**) The correlation between COMP and CAFs in pan cancers. (**F**) The correlation between COMP and CAFs in COAD and READ.

**Figure 8 biomolecules-12-01877-f008:**
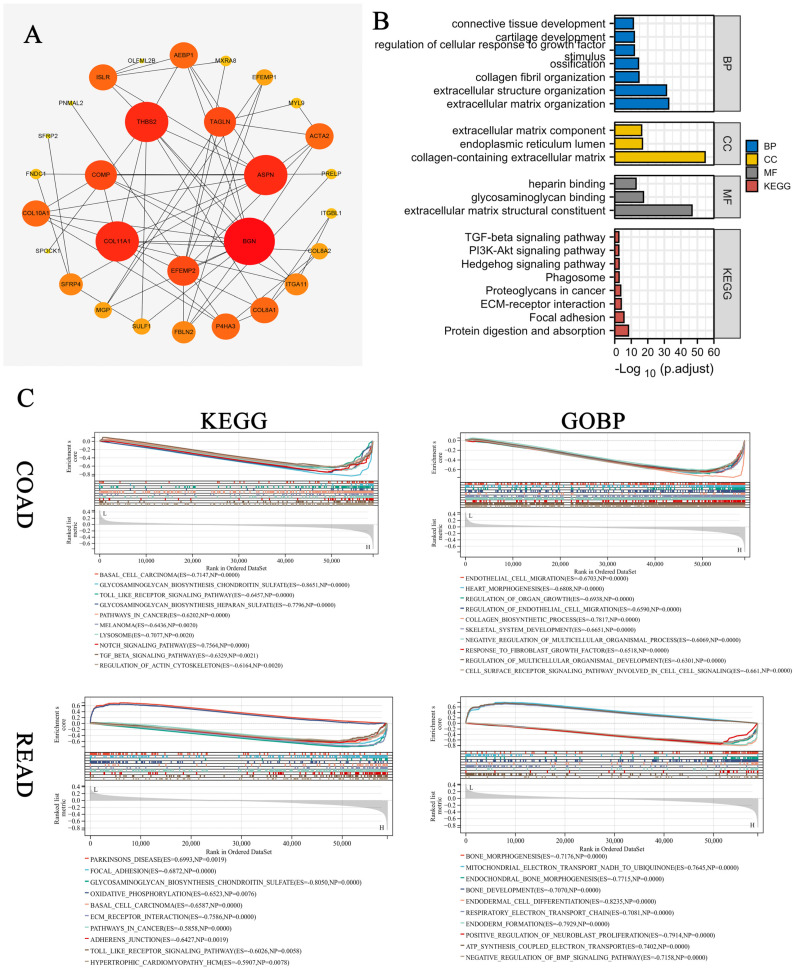
The function analysis of COMP-related genes. (**A**) The protein–protein interaction network of COMP-related genes. (**B**) GO and KRGG pathway enrichment analysis of COMP-related genes. (**C**) GSEA analysis with samples divided via median COMP expression level for COAD and READ.

**Table 1 biomolecules-12-01877-t001:** Univariable and multivariate analysis of risk factors for CRC.

		Univariate Analysis				Multivariate Analysis			
Features	Number	HR ^1^	95% CI ^1^	*p*-Value	q-Value ^2^	HR ^1^	95% CI ^1^	*p*-Value	q-Value ^2^
Gender	455			0.64	0.84				
Female		-	-						
Male		1.10	0.74, 1.64						
Age	454			**0.013**	**0.029**				
<40		-	-			-	-		
>40		0.54	0.12, 2.37			0.39	0.09, 1.75	0.2	0.3
>60		1.13	0.28, 4.64			1.32	0.32, 5.42	0.7	0.7
Molecular subtype	455			0.84	0.84				
Unstable chromatin type		-	-						
Stable genome type		1.02	0.58, 1.80						
Super mutated single-nucleotide type		0.46	0.06, 3.29						
Unstable microsatellite type		1.03	0.59, 1.81						
geographic area	360			0.79	0.84				
East Asia		-	-						
Eastern Europe		0.52	0.07, 4.05						
America		0.54	0.07, 4.00						
Western Europe		0.67	0.09, 5.05						
Organ	455			**0.028**	**0.050**				
colon		-	-	-	-				
rectum		0.55	0.31, 0.97			0.49	0.27, 0.89	**0.019**	**0.043**
Stage	443			**<0.001**	**<0.001**				
I		-	-			-	-		
II		1.62	0.67, 3.92			1.44	0.59, 3.50	0.4	0.5
III		2.75	1.16, 6.57			2.71	1.13, 6.48	0.026	0.045
IV		6.10	2.50, 14.9			7.20	2.93, 17.7	<0.001	<0.001
TGF-β subtype	455			**0.004**	**0.013**				
C1		-	-			-	-		
C2		0.54	0.36, 0.83			0.56	0.36, 0.88	0.011	0.038

^1^ HR: hazard ratio, CI: confidence interval, ^2^ Benjamini–Hochberg-adjusted *p*-value. The *p*-value ≤ 0.05 and q-value ≤ 0.05 were considered as statistically significant, which was emphasized with the bold.

## Data Availability

Not applicable.

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
