# Peer review of "TGF-β Pathways Stratify Colorectal Cancer into Two Subtypes with Distinct Cartilage Oligomeric Matrix Protein (COMP) Expression-Related Characteristics"

_biomolecules, 2022, doi:10.3390/biom12121877_

Round 1

Reviewer 1 Report

Thank you very much for this paper.  This is an interested dive into the molecular changes associated with different subgroups of colorectal cancer.

The paper will require a thorough proof-reading there are numerous typos such as “Colorectal cancers (CRC) keep the leading causes of cancer-related deaths worldwide,” in the first line of the background and “severing as a tumor suppressor,” rather than “serving as a tumor suppressor” in line 50.

The paper does an in-depth view of gene expression and does an excellent job of explaining how and why COMP may be a key factor in the behavior of TGF-B colorectal cancer.

As stated by the authors, this is preliminary work used to generate further investigations, but the work here can be used to generate future studies.

Author Response

Thank you for your careful review. We have carefully revised our manuscript and corrected all the grammatical and spelling problems we found.

Reviewer 2 Report

I remember that difference between rectal and colon adenocarcinomas is difficult to establish.  Did you find any difference concerning the grading or different histological invasiveness of these tumors? There are different types of inflammatory cells infiltrating these tumors and their significance?

Author Response

Thank you for your valuable and meaningful questions. In fact, the difference between colon and rectal cancer is an interesting by-product of our study, but we did not focus on this phenomenon in present research. Therefore, it is difficult for us to give the answer now. Thank you again for your question,  which let us realize the importance and value of this phenomenon, we will list it as one of our future research goals, and sincerely hope that our future works can better answer your questions.